# Comprehensive Analysis of the Effect of 20(*R*)-Ginsenoside Rg3 on Stroke Recovery in Rats via the Integrative miRNA–mRNA Regulatory Network

**DOI:** 10.3390/molecules27051573

**Published:** 2022-02-27

**Authors:** Rui Zhang, De-Yun Chen, Xing-Wei Luo, Yuan Yang, Xiao-Chao Zhang, Ren-Hua Yang, Peng Chen, Zhi-Qiang Shen, Bo He

**Affiliations:** 1Yunnan Key Laboratory of Pharmacology for Natural Products, School of Pharmaceutical Sciences, Kunming Medical University, Kunming 650500, China; dtyz395zr@sina.cn (R.Z.); xingweiluo@aliyun.com (X.-W.L.); yangyuanmail2@163.com (Y.Y.); 15887818479@139.com (X.-C.Z.); yangrenhua@kmmu.edu.cn (R.-H.Y.); 2Faculty of Food, Drugs and Health, Yunnan Vocational and Technical College of Agriculture, Kunming 650212, China; deyunchen@aliyun.com

**Keywords:** cerebral ischemia–reperfusion injury, 20(*R*)-ginsenoside Rg3, microRNAs, transcriptomics

## Abstract

MicroRNAs (miRNAs) are a class of small, endogenous, noncoding RNAs. Recent research has proven that miRNAs play an essential role in the occurrence and development of ischemic stroke. Our previous studies confirmed that 20(R)-ginsenosideRg3 [20(R)-Rg3] exerts beneficial effects on cerebral ischemia–reperfusion injury (CIRI), but its molecular mechanism has not been elucidated. In this study, we used high-throughput sequencing to investigate the differentially expressed miRNA and mRNA expression profiles of 20(R)-Rg3 preconditioning to ameliorate CIRI injury in rats and to reveal its potential neuroprotective molecular mechanism. The results show that 20(R)-Rg3 alleviated neurobehavioral dysfunction in MCAO/R-treated rats. Among these mRNAs, 953 mRNAs were significantly upregulated and 2602 mRNAs were downregulated in the model group versus the sham group, whereas 437 mRNAs were significantly upregulated and 35 mRNAs were downregulated in the 20(R)-Rg3 group in contrast with those in the model group. Meanwhile, the expression profile of the miRNAs showed that a total of 283 differentially expressed miRNAs were identified, of which 142 miRNAs were significantly upregulated and 141 miRNAs were downregulated in the model group compared with the sham group, whereas 34 miRNAs were differentially expressed in the 20(R)-Rg3 treatment group compared with the model group, with 28 miRNAs being significantly upregulated and six miRNAs being significantly downregulated. Furthermore, 415 (391 upregulated and 24 downregulated) differentially expressed mRNAs and 22 (17 upregulated and 5 downregulated) differentially expressed miRNAs were identified to be related to 20(R)-Rg3′s neuroprotective effect on stroke recovery. The Kyoto Encyclopedia of Genes and Genomes (KEGG) results showed that 20(R)-Rg3 could modulate multiple signaling pathways related to these differential miRNAs, such as the cGMP-PKG, cAMP and MAPK signaling pathways. This study provides new insights into the protective mechanism of 20(R)-Rg3 against CIRI, and the mechanism may be partly associated with the regulation of brain miRNA expression and its target signaling pathways.

## 1. Introduction

Ischemic stroke is an acute cerebrovascular disease that has become seriously harmful to human health due to its high mortality, disability and recurrence rates [1]. Cerebral ischemia–reperfusion injury (CIRI) is a pathological process of progressive aggravation of an ischemic injury that often occurs after the restoration of blood perfusion to a long time of ischemic brain tissue [2]. At present, the main drug for the clinical treatment of ischemic stroke is a recombinant human tissue plasminogen activator (rt-PA). However, its use has a strict time window limitation (within 4.5 h of reperfusion) [3], and excessive rt-PA can increase the risk of cerebral hemorrhage and edema [4]. Natural products, including plant extracts and their bioactive metabolites, have played an important role in the prevention and treatment of nervous system diseases [5]. Thus, it is especially urgent to discover new effective candidates with precise curative effects and few adverse reactions from natural products to treat CIRI.

Ginsenosides are the active ingredients of *Panax notoginseng*, which has a long application history in East Asian nations such as China, Korea and Japan. It has various pharmacological effects, such as antiaging, anti-inflammatory and antitumor effects [6,7,8]. 20(*R*)-ginsenoside Rg3 [20(*R*)-Rg3] is a monomeric compound of ginsenosides that is extracted and isolated from the traditional Chinese medicine *Panax notoginseng* (Figure 1). Liu et al. [9] found that 20(*R*)-Rg3 can significantly improve oxidative stress of the SK-N-SH neuroblastoma cells induced by H_2_O_2_. Our previous studies have confirmed that treatment with 20(*R*)-Rg3 can improve neurological function, reduce the cerebral infarction volume and inhibit the apoptosis of neural cells in a middle cerebral artery occlusion and reperfusion (MCAO/R) rat model and oxygen-glucose deprivation/reperfusion (OGD/R) model using SH-SY5Y cells [10,11]. Although the neuroprotective effect of 20(*R*)-Rg3 on CIRI has been confirmed, the effect of subsequent 20(*R*)-Rg3 therapy on dysregulated microRNAs (miRNAs) has not yet been fully elaborated.

MiRNAs are short non-coding RNAs with a length of 21–23 base pairs (bp) that work as regulators of the expression of mRNA [12,13]. Studies have shown that a variety of miRNAs participate in the regulation of CIRI by affecting multiple signaling pathways, such as neuronal apoptosis, the inflammatory response and oxidative stress [14,15]. It has been reported that miRNA-589 can be downregulated after CIRI and that the overexpression of miRNA-589 reduces the inflammatory response by negatively regulating tumor necrosis factor receptor-associated factor 6 (TRAF6) [16]. In addition, there are also studies that indicate that the inhibitory effect of 20(*R*)-Rg3 on angiogenesis is likely related to the upregulation of miRNA-520h expression [17]. However, there has been no report on the mechanism of 20(*R*)-Rg3 treatment in attenuating CIRI in rats via integrative miRNA–mRNA regulatory network analysis.

Transcriptomics is an emerging discipline focused on the regulation at the genomic and/or transcriptomic levels containing cells or tissues from animal to human and reveals the mechanism of gene expression regulation by studying the changes in the expression of all RNA, including miRNA [18]. In this study, we applied a high-throughput sequencing approach to study the expression profiles of differential miRNAs and mRNAs in nine rat brain tissues from sham, MCAO/R- and 20(*R*)-Rg3-treated rats, and we also predicted the miRNA target genes and combined them with bioinformatics technology to analyze the biological function of target genes and possible signaling pathways. In this study, miRNAs were determined as the cutoff point to explore the protective effect of 20(*R*)-Rg3 in an MCAO/R model, and these novel findings may provide a theoretical and experimental basis for revealing the potential molecular mechanism of 20(*R*)-Rg3 against CIRI.

## 2. Materials and Methods

### 2.1. Drugs and Reagents

20(*R*)-Rg3 (purity > 99%) was generously supplied by Professor Cheng Zou (Kunming Medical University, Kunming, China). 20(*R*)-Rg3 was dissolved in dimethyl sulfoxide (DMSO), mixed with 10% Tween 80 for solubilization and diluted to 20 mg/mL with normal saline. TRIzol reagent was obtained from Ambion (Austin, TX, USA). An All-in-one™ miRNA QRT-PCR Detection Kit (QP015) was bought from iGene Biotechnology Co., Ltd. (Guangzhou, China). The primers were synthesized by Wuhan Qingke Biotechnology Co., Ltd. (Wuhan, China). The other reagents and chemicals were of analytical grade.

### 2.2. Animals and Grouping

In order to reduce animal mortality and ensure the stability and reliability of the MCAO/R model, we selected rats with a body weight of 280–320 g [20]. Male grade SPF Sprague–Dawley rats were purchased from the Laboratory Animal Center of Kunming Medical University (license number: SCXK (Yunnan) k2015-0002, Yunnan, China). Rats were housed in a controlled environment maintained at 21–23 °C and 55% humidity. All experimental protocols were approved by the Animal Care and Welfare Committee of Kunming Medical University (Yunnan, China) and were performed in strict accordance with the relevant Guidelines of the China Council on Animal Care and Use. In this study, 27 rats were randomly divided into three groups as follows (*n* = 9): the sham group, MCAO/R group and 20(*R*)-Rg3 group.

Based on our previous study and the results of the preliminary experiment, we chose to administer 20 mg/kg of 20(*R*)-Rg3 in this study [10]. In order to better assess the preventive and therapeutic effects of 20(*R*)-Rg3 and explore its potential molecular mechanism, 20(*R*)-Rg3 was administered three times by intraperitoneal (i.p.) injection, and the dosing regimen was as follows: the first dose of 20(*R*)-Rg3 was given 12 h before surgery, followed the onset of reperfusion by the second dose, and the third dose was immediately administered 12 h after reperfusion to these rats.

### 2.3. MCAO/R Model

The MCAO/R rat model was replicated by referring to the method reported by Bo He et al. [10]. Briefly, the rats were anesthetized using 2% isoflurane and fixed on the surgical table, and the cervical hair was shaved with an electric razor. Then, a small incision of approximately 3.0 cm was made along the medioventral line. The right common carotid artery (CCA), internal carotid artery (ICA) and external carotid artery (ECA) were bluntly exposed, and the distal portion of the ECA was ligated. A 0.24-mm nylon thread was inserted into the middle cerebral artery at a depth of approximately 20 mm. The incision was sutured, and the nylon suture was fixed firmly. The temperature of the rats was maintained at 36.5 ± 0.5 °C with a heating lamp. After 2 h of ischemia, the nylon thread was gently pulled out to allow reperfusion.

### 2.4. Neural Behavioral Test

At 24 h of reperfusion, the neurobehavioral score was assessed according to Longa’s test [21]. Briefly, the scoring criteria were as follows: 0, no observable deficit; 1, partial neurologic impairment (endoduction of the contralateral forelimb and not wholly stretched); 2, circling spontaneously to the contralateral side; 3, inability to contralaterally side and 4, without spontaneous walking and some consciousness lost.

### 2.5. Open-Field Experiment

The open-field arena (100 cm × 100 cm × 40 cm) was placed in a dimly lit (40 lux) room to reduce external interference. After the neural behavioral test, the rats were put into the open-field arena and allowed to adapt to the environment for 5 min, and the movement track, movement distance and average movement rate within 15 min were video recorded and measured. To avoid animal odor interference, the chamber was swabbed with 10% alcohol cotton after testing each animal.

### 2.6. Hematoxylin–Eosin (HE) Staining

HE staining was performed to assay the pathological changes in the ischemic penumbra. After the neural behavioral test and open-field experiment, the rats were euthanized. The rats were perfused with 0.9% normal saline and 4% paraformaldehyde through the heart, and the brain tissue was embedded in paraffin and sliced into 4-μm-thick sections for HE staining.

### 2.7. Sequencing Sample Preparation

Due to the limitations of the research funds and to ensure statistical significance with the lowest number of animals, we randomly selected three rats in each group for the sequencing experiment. The cerebral cortex was collected from three rats in each group, and a 2 mm × 2 mm × 2 mm section from the ipsilateral ischemic penumbra was isolated, placed into TRIzol reagent and stored in the refrigerator at −80 °C for sequencing.

### 2.8. miRNA Microarray Analysis

The total RNA of each sample was isolated with TRIzol reagent (Ambion, Austin, TX, USA). A NanoPhotometer spectrophotometer (IMPLEN, Westlake Village, CA, USA) was used to check the purity of the RNA. The Qubit B RNA Assay Kit on a Qubit 2.0 fluorometer (Life Technologies, New York, NY, USA) was used to test the concentration of RNA. In addition, in order to evaluate RNA integrity, we used the RNA Nano 6000 Assay Kit in the Bioanalyzer 2100 system (Agilent Technologies, Santa Clara, CA, USA). After passing the total RNA detection of the samples, the RNA libraries were prepared with a small RNA Sample Prep Kit (Illumina, San Diego, CA, USA).

### 2.9. RT-PCR

We randomly selected 6 differentially expressed miRNAs and 6 differentially expressed mRNAs for RT-PCR verification, and the primer sequences are shown in Appendix A. The All-In-One™ miRNA QRT-PCR Detection Kit was used to detect the miRNA levels in the samples. RT-PCR was performed using a real-time PCR system (ABI7500; Applied Biosystems, Carlsbad, CA, USA). The expression levels of the miRNA were normalized to U6, and the relative quantities of the mRNA levels were normalized to glyceraldehyde-3-phosphate dehydrogenase (GAPDH), which were calculated by the 2^−ΔΔCt^ method [22].

### 2.10. Prediction of miRNA-Targeted Genes and Construction of the miRNA–mRNA Network

To predict the candidate target genes of miRNAs, we chose the TargetScan (http://www.targetscan.org/, Cambridge, MA, USA. Accessed on 27 July 2021) and miRanda (http://www.miranda.org/, Philmont, NM, USA. Accessed on 27 July 2021) databases. The screening criteria were applied: correlation > 0.99 and *p* < 0.05. We selected those genes that overlapped in both databases for the further functional analysis. Then, the functional annotation of these genes of the differentially expressed miRNAs was performed with DAVID v6.7. The miRNA–mRNA interaction network was constructed by using Cytoscape software (v 3.5.2, Bethesda, MD, USA).

### 2.11. GO Analysis and KEGG Pathway Analysis

To further explore the functional roles of the differentially expressed protein-coding genes, we performed Gene Ontology (GO) [23] category and Kyoto Encyclopedia of Genes and Genomes (KEGG) [24] pathway analyses. The top twenty most significantly enriched GO terms and KEGG pathways are presented.

### 2.12. Statistical Analysis

SPSS 18.0 software (IBM Corp., Chicago, IL, USA) was used for the analyses. All results were expressed as the mean ± standard deviation. The two groups were compared using a two-tailed Student’s *t*-test. Differential miRNAs were analyzed using the DEGseq (2010) R package. The *p*-value was corrected using the *q*-value according to the method adopted by Story [25]. A *p*-value less than 0.05 was considered statistically significantly different.

## 3. Results

### 3.1. 20(R)-Rg3 Can Improve Neurobehavioral Dysfunction in MCAO/R-Treated Rats

The open-field test and Longa neurological score were employed to observe the neurobehavioral changes, and the results are shown in Figure 2. Compared with the sham group, the total distance, the average speed of movement and the neurological score were significantly decreased in the model group, while these changes were significantly increased in comparison to the 20(*R*)-Rg3 group (*p* < 0.05 and *p* < 0.01, Figure 2A–D). The HE staining results showed that, compared with neurocytes in the sham group, neurocytes in the model group displayed a sparse and disordered arrangement with a blurred structure. Neurocytes treated with 20(*R*)-Rg3 showed a relatively intact structure with an orderly arrangement (Figure 2E). These results suggest that 20(*R*)-Rg3 can significantly improve MCAO/R-induced brain injury and neurobehavioral dysfunction in rats.

### 3.2. Overview of mRNA and miRNA Sequencing

Through mRNA sequencing, we obtained a total of 859,304,654 raw reads (241,687,218 for the sham group, 316,916,154 for the model group and 300,701,282 for the 20(*R*)-Rg3 group). By comparing the uniquely mapped reads, a total of 669,582,916 data points to be analyzed were found (213,951,465 for the sham group, 232,780,986 for the model group and 222,850,465 for the 20(*R*)-Rg3 group). The mapping rates of the sham group were 77.97%, 79.74% and 79.40%; those of the model group were 74.72%, 73.69% and 71.98% and those of 20(*R*)-Rg3 were 88.17%, 88.72% and 88.68%, respectively.

For miRNA sequencing, we generated 132,130,113 raw reads (32,216,409 for the sham group, 51,127,624 for the model group and 48,786,080 for the 20(*R*)-Rg3 group). After discarding the reads with low quality (the base number of SQ < = 20 accounts for more than 30% of the whole read), a total of 130,475,920 data to be analyzed were retained (31,725,247 for the sham operation group, 50,453,567 for the model group and 48,297,106 for the 20(*R*)-Rg3 group). The mapping rates of the sham group were 77.97%, 79.74% and 79.40%; those of the model group were 70.22%, 82.09% and 77.37% and those of 20(*R*)-Rg3 were 70.93%, 65.17% and 72.80%, respectively.

We screened miRNAs from the data to be analyzed of each sample and matched them with the reference sequence. A total of 685 known mature miRNAs were detected, and 111 novel mature miRNAs and 111 miRNA precursors were revealed.

### 3.3. Cluster Analysis of Differentially Expressed mRNAs and miRNAs

We performed an integrated analysis of the mRNA and miRNA expression profiles, and the results were visualized by hierarchical clustering heatmaps (Figure 3A), volcano plots and Venn diagrams. We set a log2-fold change ≥ 1 and false discovery rate < 0.05 as the thresholds to reduce the false-positive rate. Compared with the sham group, a total of 3555 mRNAs were differentially expressed in the model group, including 953 upregulated mRNAs and 2602 downregulated mRNAs (Figure 3B). Compared with the model group, 472 mRNAs were differentially expressed in the 20(*R*)-Rg3 group. Among them, 437 mRNAs were upregulated, and 35 mRNAs were downregulated (Figure 3C). The mRNA sequencing results are shown in Appendix A. To further analyze the expression profiles of differential mRNAs, we selected the intersection of 953 upregulated mRNAs in the model group and 35 downregulated mRNAs in the 20(*R*)-Rg3 group to obtain 24 differentially expressed mRNAs (Figure 3D); 2602 downregulated mRNAs in the model group and 437 upregulated mRNAs in the 20(*R*)-Rg3 group were used to collect 391 differentially expressed mRNAs (Figure 3E).

The differential expression miRNAs were visualized by hierarchical clustering heatmaps (Figure 4A). Compared with the sham group, there were 283 miRNA changes in the model group, including 142 upregulated miRNAs and 141 downregulated miRNAs (Figure 4B). It was found that there were 34 differentially expressed miRNAs in the 20(*R*)-Rg3 group compared to the model group: six miRNAs were downregulated, and 28 miRNAs were upregulated (Figure 4C). The miRNA sequencing results are shown in Appendix A. We also created a Venn diagram to show the intersection among the three groups for the expression profiles of differential miRNAs. A Venn diagram showed that there were five intersection miRNAs from 142 upregulated miRNAs in the model group and six downregulated miRNAs in the 20(*R*)-Rg3 group (Figure 4D) and 17 intersection miRNAs from 141 downregulated miRNAs in the model group and 28 upregulated miRNAs in the 20(*R*)-Rg3 group (Figure 4E). It is suggested that 20(*R*)-Rg3 treatment could reverse, to a large extent, these transcriptomic alterations induced by MCAO/R.

### 3.4. Validation of the Sequencing Data by Quantitative RT-PCR Analysis

We randomly selected and confirmed the expression of six mRNAs and six miRNAs with significant expression differences by qRT-PCR to confirm the accuracy of the sequencing results. The results showed that, compared with the sham group, the expression trends of mRNA and miRNA in the model group were consistent with the sequencing results (Figure 5). Meanwhile, compared with the model group, the expressions of the above mRNAs and miRNAs in the 20(*R*)-Rg3 group showed a similar trend to the sequencing results. Thus, the results of the qRT-PCR were in good agreement with the RNA sequencing data.

### 3.5. GO Analysis and KEGG Pathway Enrichment Analysis

To understand the function of differentially expressed miRNAs, we assessed the predicted molecular function and pathway by using RNAhybrid and miRanda software. GO function enrichment items included three aspects: biological process (BP), cellular component (CC) and molecular function (MF). The GO items of the target genes in the sham group and the model group are shown in Figure 6A; in the BP, the response to external stimulus (GO: 0009605), informational response (GO: 0006954) and defense response (GO: 0006952) were the top three GO terms; in the CC, the extracellular region (GO: 0005576), extracellular space (GO: 0005615) and extracellular region part (GO: 0044421) were the top three GO terms and in the MF, protein binding (GO: 0005515), protein-containing complex binding (GO: 0044877) and binding (GO: 0005488) were the top three GO terms. The GO items of the target genes in the model group and the 20(*R*)-Rg3 group are shown in Figure 6B; in the BP, positive regulation of the biological process (GO:0048518) and regulation of the response to stimulus (GO:0048583) and response of stress (GO:0006950) were the top three GO terms; in the CC, the extracellular region (GO:0005576), extracellular space (GO:0005615) and plasma membrane part (GO:0044459) were the top three GO terms and in the MF, binding (GO:0005488), signaling receptor binding (GO:0005102) and protein binding (GO:0005515) were the top three GO terms. The top 20 differentially enriched pathways identified by KEGG pathway analysis are shown in Figure 6C and were mainly enriched in the ECM–receptor interaction and cGMP-PKG, cAMP and MAPK signaling pathways.

### 3.6. Construction of the miRNA-Target Regulation Network

We analyzed the predicted and validated differentially expressed miRNAs and target genes, counted the miRNAs and mRNAs with a degree of more than 20 and constructed the miRNA–target gene regulation network by using Cytoscape software. As shown in Figure 7, 10 out of the predicted miRNAs were chosen after taking the intersection of the predicted genes. Then, it was observed that 10 differentially expressed miRNAs could target 85 differentially expressed target genes from three different databases. These results indicated that 20(*R*)-Rg3 may exert neuroprotective effects against CIRI by modulating the expression of multiple differentially expressed miRNAs and their target genes.

## 4. Discussion

Cerebral ischemia–reperfusion injury is caused by the restoration of blood supply after cerebral tissue ischemia for a certain period of time. A series of pathophysiological changes occur following cerebral ischemia–reperfusion, including disturbance of the ion balance and energy metabolism, inflammation, excitatory amino acid toxicity, oxidative stress and apoptosis [26]. The method of MCAO/R is a classical model of CIRI that can simulate human stroke with good stability and high repeatability [27]. Previous studies have reported that the monomer compound 20(*R*)-Rg3 has anti-inflammatory and antioxidant effects. Li et al. [28] found that 20(*R*)-Rg3 significantly inhibited the oxidative stress induced by d-galactose in the liver and kidneys, increased the levels of catalase (CAT) and superoxide dismutase (SOD) and reduced the contents of malondialdehyde (MDA) and 4-hydroxynonenal (4-HNE). Its mechanism may be related to the activation of the PI3K/Akt signaling pathway. Ahn et al. [29] found that 20(*R*)-Rg3 can reduce the expression of tumor necrosis factor-alpha (TNF-α), interleukin-6 (IL-6) and inducible nitric oxide synthase (iNOS) in Aβ-induced microglia and play a neuroprotective role by inhibiting the inflammatory response. Yoon et al. [30] demonstrated that 20(*R*)-Rg3 could suppress both lethal endotoxic shock and S-nitrosylation of the NLRP3 inflammasome by inhibiting nitric oxide (NO) production through the regulation of iNOS expression. These studies suggest that 20(*R*)-Rg3 may serve as a candidate therapeutic agent for both inflammatory and oxidative stress-related diseases. Many studies on the potential mechanism of action of 20(*R*)-Rg3 are limited to a single target and/or pathway, which fails to reflect the “multiple target and multiple pathway” neuroprotective effect. In this study, we performed high-throughput sequencing to explore the differentially expressed genes and miRNAs of 20(*R*)-Rg3 and offers a molecular basis to further understand the protective mechanism of 20(*R*)-Rg3 in the treatment of CIRI.

First, the treatment of 20(*R*)-Rg3 significantly ameliorated neurobehavioral impairment in MCAO/R-induced rats, measured by using neurobehavioral scores and the open-field test. Then, in this study, we found that neurocytes in the model group appeared in a sparse and disordered arrangement with a blurred structure. By contrast, in the 20(R)-Rg3 group, the neurocytes displayed a relatively intact structure with an orderly arrangement. It has been reported that MCAO/R induced ischemic neuron injury and microglial activation [31]. The activation of microglia is a key factor in neuron loss, and attention to the microglia can provide an opportunity to intervene with drugs or genetic markers for post-stroke treatment [32,33]. Previous studies have confirmed that 20(*R*)-Rg3 can inhibit microglial responses and play a neuroprotective role at the cellular level [29,34]. In this study, we confirmed the protective effect of 20(*R*)-Rg3 on neurons in the MCAO/R model, but the effect of 20(*R*)-Rg3 on the microglia will be further explored in our future work.

In addition, we used the Illumina HiSeq TM2500 platform to sequence the brain tissue from the cerebral infarct area in the sham, MCAO/R and 20(*R*)-Rg3 groups, and a total of 669,582,916 mRNAs and 130,475,920 miRNAs were obtained. Our results showed that there were 3555 (953 upregulation and 2602 downregulation) differentially expressed mRNAs between the sham group and model group and 472 (437 upregulation and 35 downregulation) differentially expressed genes between the model group and 20(*R*)-Rg3 group. In all, 283 (142 upregulated and 141 downregulated) differentially expressed miRNAs were found between the sham group and the model group, and 34 (28 upregulated and six downregulated) differentially expressed miRNAs were obtained between the model group and 20(*R*)-Rg3 group. The Venn diagrams showed that 20(*R*)-Rg3 treatment regulates 415 (391 up- and 24 downregulated) differentially expressed mRNAs and 22 (17 up- and five downregulated) differentially expressed miRNAs in MCAO/R-treated rats. Furthermore, the GO functional analysis and KEGG enrichment analysis were performed to explore the target genes of differentially expressed mRNAs and miRNAs. The miRNA–mRNA network diagram was constructed to explore the effect and underlying molecular mechanism of the protective role of 20(*R*)-Rg3 against CIRI.

It has been reported that the expression of multiple genes/pathways is regulated after CIRI. Jeyaseelan et al. [35] performed a miRNA sequencing analysis to identify differentially expressed genes between normal and MCAO rats and found that the expression of these genes, such as transgelin (TAGLN), vimentin (VIM) and prostaglandin E synthase (PTGES), was significantly altered. The results obtained by Meng et al. [36] showed that 817 differentially expressed mRNAs were related to the neuroprotective effect of *Panax notoginseng* on stroke recovery by sequencing. Liu et al. [37] investigated the protective effect of β-cyanide on CIRI in rats by using mRNA sequencing and found 411 differentially expressed mRNAs—among which, the Pax1, Cxcl3 and Ccl20 genes were significantly altered. In our study, 415 mRNAs were found to be associated with the effect of 20(*R*)-Rg3 on stroke recovery, while the expression of these genes, such as Tagln, Vim, Ptges, Pax1, Cxcl3 and Ccl20, was also significantly altered in our sequencing results. In the present study, we found that 20(*R*)-Rg3 may play a protective role against CIRI through the targeting of multiple mRNAs.

Studies have confirmed that changes in the miRNA expression profile can be used as potential diagnostic and therapeutic biomarkers of nervous system-related diseases [38]. Duan et al. [39] used the high-throughput sequencing method to detect the changes in differentially expressed miRNAs between normal and MCAO/R rats and found that, compared with the normal group, 37 differentially expressed miRNAs in the MCAO/R group were increased, such as miR-155-5p, miR-223-3p and miR-183-5p. Zou et al. [40] constructed a miRNA–mRNA expression network through sequencing to explore the neuroprotective effect of biliverdin on CIRI in rats and identified 81 miRNAs related to the protective effect of biliverdin against CIRI. In this study, we found that 283 (142 up- and 141 downregulated) miRNAs were associated with CIRI in the MCAO/R group compared with the sham group, and 34 (28 up- and six downregulated) miRNAs were related to the neuroprotective effect of 20(*R*)-Rg3 for the treatment of CIRI. In this study, we showed that 20(*R*)-Rg3 may play a protective role against CIRI by regulating multiple miRNAs, and the underlying mechanism remains to be further explored.

GO analysis can gain an understanding of the biological properties of target genes and clarify the changes in gene functions through sample differences in the experiment [41,42], which can more accurately calculate the probability of GO items enriched for different genes [43]. Duan et al. [44] conducted a GO enrichment analysis on the differentially expressed genes in normal and MCAO/R rats, suggesting that the significantly highly enriched CC items in the model group included the cell surface, vesicles and cytoplasmic cells, and the most enriched MF item was protein binding. In the present study, we found that the enriched GO terms in the miRNA target genes were protein binding, cell surface, vesicles and cytoplasmic cells.

KEGG analysis is a primary public database of signaling pathways that can be used to find key genes related to diseases [45]. The pathological process of CIRI is complex, and it has been confirmed that a variety of miRNAs are involved in the regulation of signal pathway transduction in the occurrence and development of CIRI [46,47,48]. Yuen et al. [49] showed that erythropoietin and cyclosporine treatment can mitigate ischemic brain damage by inhibiting inflammation after acute ischemic stroke in rats, and its molecular mechanism was principally associated with the MAPK signaling pathway. Gao et al. [50] found that ginsenoside Rb1 can promote neurotransmitter release and improve functional recovery after stroke via the cAMP pathway. Meng et al. [36] revealed that Panax notoginseng has a protective effect on cerebral ischemia and that its possible molecular mechanisms involve the Rap1, cAMP, and cGMP-PKG signaling pathways. Our results showed that these pathways were potentially related to the neuroprotective effect of 20(*R*)-Rg3, including the cGMP-PKG, cAMP and MAPK signaling pathways, and the specific mechanisms of action and regulation of these signaling pathways warrant further investigation.

It has been reported that the inhibition of miR-19a-3p can alleviate CIRI by improving inflammation and apoptosis in vivo and in vitro [51]. Wang et al. [52] found that ginsenoside Rb1 can increase the expression of miR-130b-5p in the spinal cord tissues of rats after spinal cord injury and decrease the expression of inflammatory factors. Mota et al. [53] found that resveratrol treatment can reduce the expression of Tgfb1 and inflammatory factors in MCAO/R mice. Zhang et al. [54] found that treatment with RT-PA and 2-(2-Benzofu-Ranyl)-2-imidazoline could alleviate brain injury induced by delayed thrombolysis and reduce the expression of intercellular adhesion molecule 1 (ICAM1). In the present study, we constructed a miRNA–mRNA interaction network diagram and revealed that multiple differentially expressed miRNAs (miR-204-3p, miR-449a-5p and miR-21-3p) could regulate the expression of target genes such as Tgfb1, ICAM1 and GFAP. Nevertheless, the interaction mechanisms of miRNA–mRNA that underlie 20(*R*)-Rg3 neuroprotection deserve further study.

## 5. Conclusions

In summary, our results suggest that 20(*R*)-Rg3 can improve neurobehavioral dysfunction in rats following MCAO/R. The molecular mechanism may be related to changes in the cGMP-PKG, cAMP and MAPK signaling pathways. These differentially expressed miRNAs, genes and pathways could serve as potential targets of therapy for further research on the neuroprotective effect of 20(*R*)-Rg3 in ischemic stroke.

## Figures and Tables

**Figure 1 molecules-27-01573-f001:**
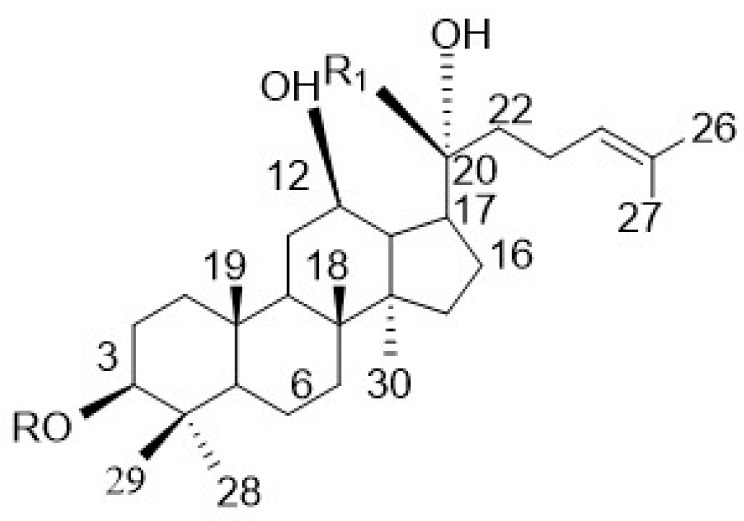
Chemical structure of 20(*R*)-ginsenoside Rg3. R: -glc(2 → 1)glc (glc: b-D-glucopyranosyl); R_1_: -CH_3_ [19].

**Figure 2 molecules-27-01573-f002:**
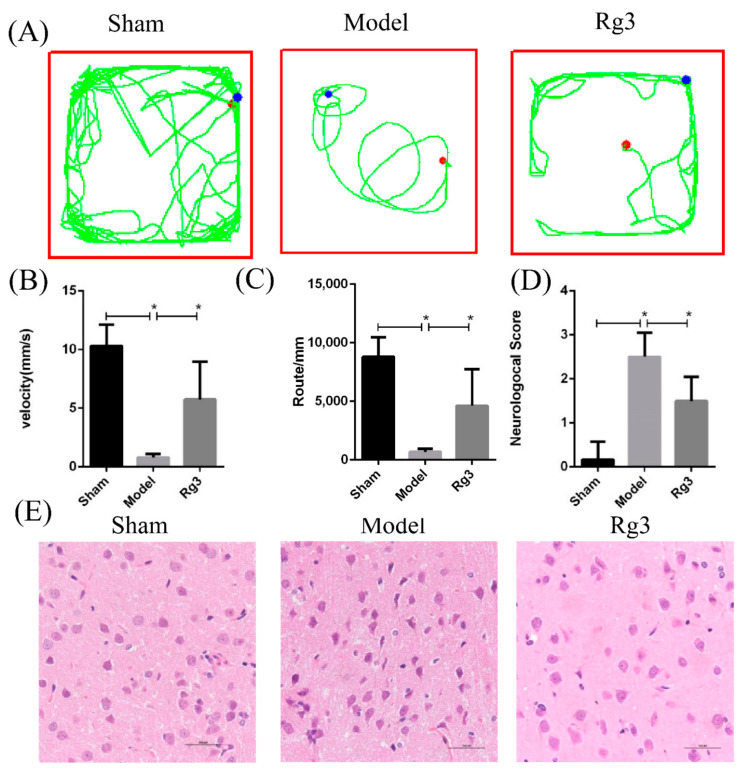
Assessment of neurobehavior following 20(*R*)-Rg3 treatment in rats with MCAO/R injury. The open-field test and neurobehavioral scoring were performed in rats that underwent cerebral ischemia-reperfusion injury (2 h/24 h). (**A**) Track plot of the rats during the open-field test. (**B**) The average speed of movement traveled as measured with the open-field test. (**C**) The total distance traveled in the open-field test. (**D**) Statistical analysis of the Longa’s test scores. (**E**) Representative images of brain tissue in rats by HE staining (scale bar = 100 μm). Data are represented as the mean  ±  SD, * *p* < 0.05, *n* = 9 per group.

**Figure 3 molecules-27-01573-f003:**
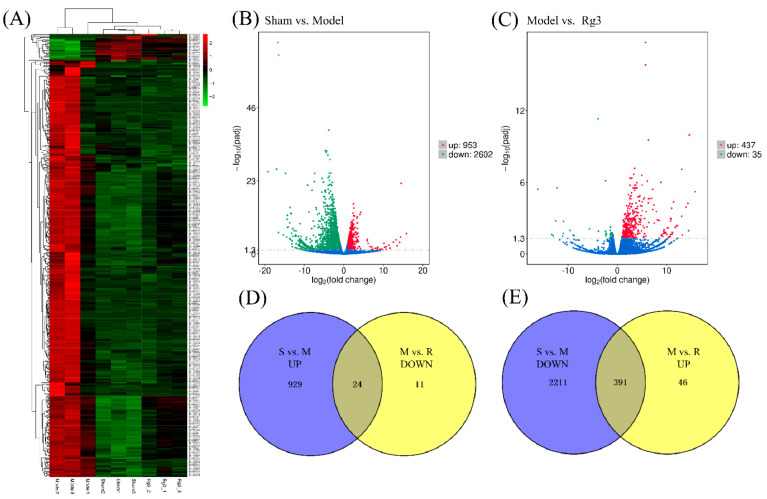
Screening for differential mRNA expression. (**A**) Hierarchical clustering heatmap of the differential mRNA expression. Blue represents downregulated mRNA, red represents upregulated mRNA and the darker the color, the more significant the mRNA difference. (**B**,**C**) Volcano plots of the differential mRNA expression. (**D**,**E**) Venn diagrams of the differential mRNA expression.

**Figure 4 molecules-27-01573-f004:**
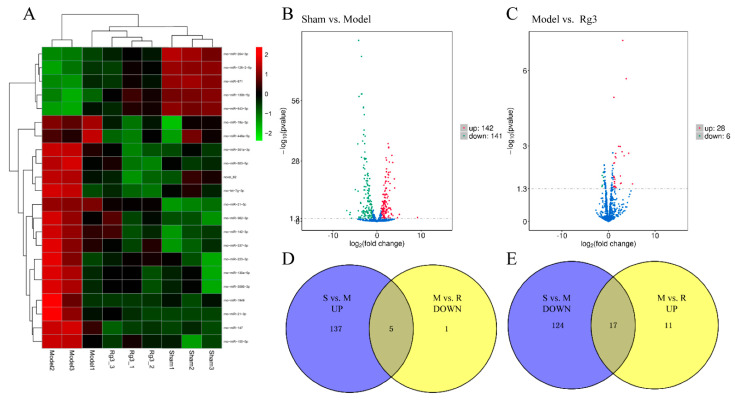
Screening for differential miRNA expression. (**A**) Hierarchical clustering heatmap of the differential miRNA expression. Blue represents downregulated miRNAs, red represents upregulated miRNAs and the darker the color, the more significant the difference between miRNAs. (**B**,**C**) Volcano plots of the differential miRNA expression. (**D**,**E**) Venn diagrams of the differential miRNA expression.

**Figure 5 molecules-27-01573-f005:**
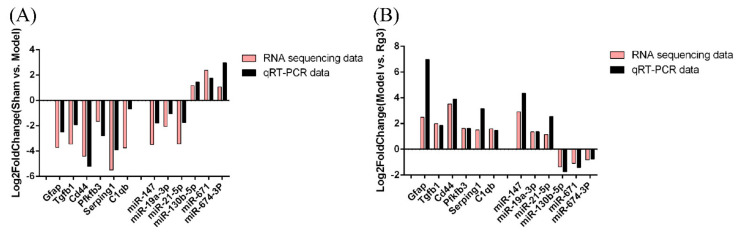
Validation of the transcript expression of the mRNAs and miRNAs by real-time quantitative polymerase chain reaction. (**A**) Differentially expressed mRNAs and miRNAs in the sham group and model group were randomly selected. (**B**) Differentially expressed mRNAs and miRNAs in the model group and 20^®^-Rg3 group were randomly selected.

**Figure 6 molecules-27-01573-f006:**
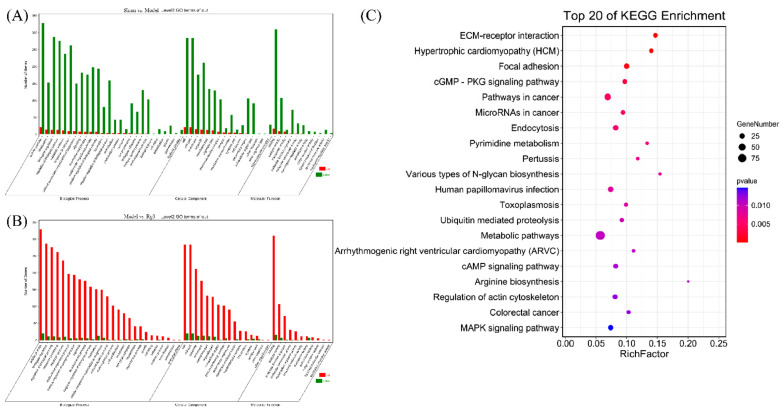
GO and KEGG analysis of the candidate miRNA target genes. (**A**) GO enrichment analysis of the candidate miRNA target genes in the sham group and the model group. (**B**) GO enrichment analysis of the candidate miRNA target genes in the model group and the 20(*R*)-Rg3 group. (**C**) KEGG pathway enrichment. The vertical axis represents different pathways, and the horizontal axis represents the proportion of the differentially expressed genes. The redder color of the dots represents more significant enrichment. The size of the circle indicates the number of genes enriched in the pathway. GO, gene ontology. KEGG, Kyoto Encyclopedia of Genes and Genomes. BP, biological process. CC, cellular component. MF, molecular function.

**Figure 7 molecules-27-01573-f007:**
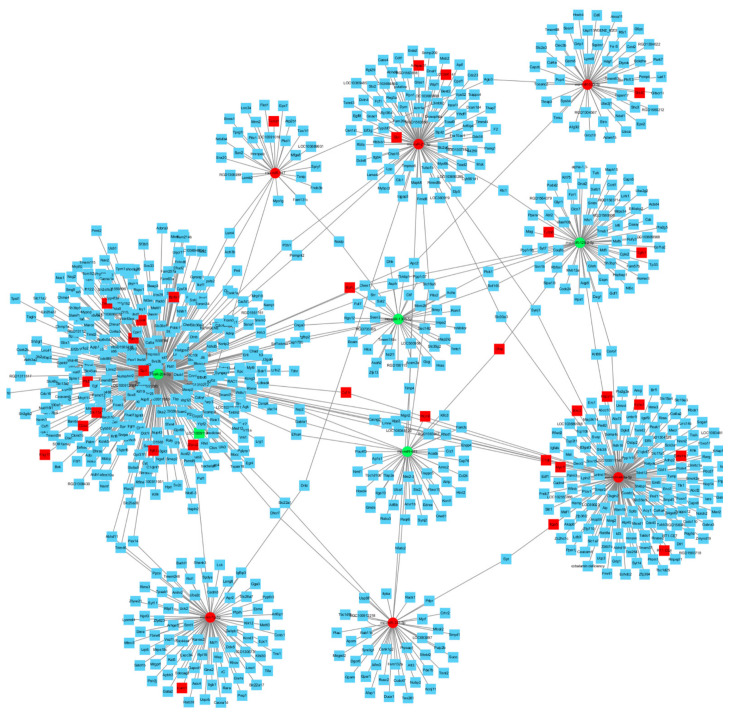
The miRNA–mRNA interaction network diagram. Circles and squares represent miRNAs and mRNAs, respectively. Green represents downregulation, and red represents upregulation. Blue represents the predicted target genes.

## Data Availability

Not applicable.

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
