# Peer review of "Comprehensive Analysis of the Effect of 20(R)-Ginsenoside Rg3 on Stroke Recovery in Rats via the Integrative miRNA–mRNA Regulatory Network"

_molecules, 2022, doi:10.3390/molecules27051573_

Round 1

Reviewer 1 Report

Authors used an animal experimental study to identify the effect of 20(R)-Rg3 on recovery of ischemic stroke. Because ischemic stroke is one of the most popular diseases with high mortality, it is important to develop therapeutic drugs. However, this article has not been fully answered some of questions due to the insufficient description.

First, authors used 20mg/kg of 20(R)-Rg3 (L115), but they did not explain why they used this dose. Authors should justify why they used 20 mg/kg of 20(R)-Rg3 in method section.

Second, authors used SD rats weighing approximately 280-320, but they did not explain why they used SD rats and why they used a rat of this weight. Authors should justify them in method section.

Third, authors explained that the first dose of 20(R)-Rg3 was given 12 h “before” surgery (L117). However, if authors aim to use 20(R)-Rg3 for treatment but not for prevention, it may be important to use it “after” surgery (i.e., after the attach of ischemic stroke). Authors should justify why they used 20(R)-Rg3 before surgery.

Fourth, authors explained that they did 3 things at 24 h of reperfusion (i.e., neural behavioral test in L132, open-field experiment in L139 and HE staining in L146), but it may not be possible to do all three things at the same time. Authors should justify it in method section.

Finally, authors used three randomly selected rats but not all rats for collection of cerebral cortex. Authors should justify why they did not use all rats in method section.

Minor comments

  1. Some of abbreviations in abstract should be spelled out (e.g., SD in L21 and KEGG in L39).
  2. Reference should be cited in the following text "the expression levels ... by the 2-ddCt method" in L167-L169 and "we performed Gene Ontology (GO) category .... are presented." in L180-L182.
  3. It is difficult to read figure 6 and figure 7, because the letters were too small to read.

Reviewer 2 Report

In this paper, the authors cited their previous studies in that they showed as the treatment with 20(R)-ginsenosideRg3 [20(R)-Rg3] can improve neurological function, reduce cerebral infarction volume and inhibit apoptosis of neural cells in the middle cerebral artery occlusion and reperfusion (MCAO/R) rat model.

Starting from these results, in this paper they used high-throughput sequencing to investigate the differentially expressed miRNA and mRNA expression profiles of 20(R)-Rg3 preconditioning to ameliorate CIRI injury in rats and to reveal its potential neuroprotective molecular mechanism.

The authors claim “Previous research has established that the monomer compound 20(R)-Rg3 has anti-inflammatory and neuroprotective effects on central nervous system diseases, but related studies are limited to a single target and/or pathway, which fails to reflect the “multiple-target and multiple- pathway” neuroprotective effect. So, their intent is to highlight in a "comprehensive" way the changes that occur in animals treated with 20(R)-Rg3 compared to untreated model. Although this work seems to have a very wide methodological part and, in some respects, also innovative, in my opinion it doesn't fully reflect what the reader expects reading the title of the work.

    • In the title, the word "identification" in my opinion is too ambitious compared to the results of the work and for me it should be changed.

    • It would be much appreciated if the authors implemented in “Introduction” and/or “Discussion” the actual knowledges about the mechanism of action of 20(R)-Rg3. They cite references 22 and 23, but it would be interesting to understand the current state of knowledge about the mechanism of action of 20(R)-Rg3 and how the results of their work fit into this context.

    • In Figure 2E, the immunohistochemistry data are not interpreted and discussed. Authors should interpret the images and perhaps implement them. In fact, it is known that MCAO/R model induces both neuronal and microglia changes. In this animal model, relationship betwen activated microglia and 20(R)-Rg3 is already known as the same authors have mentioned with riferiment 23. The authors should interpret their results taking into account these important knowledges and also cite the following riferiments:

    -Morioka T, Kalehua AN, Streit WJ. Characterization of microglial reaction after middle cerebral artery occlusion in rat brain. J Comp Neurol. 1993 Jan 1;327(1):123-32. doi: 10.1002/cne.903270110. PMID: 8432904.

    -Miyajima N, Ito M, Rokugawa T, Iimori H, Momosaki S, Omachi S, Shimosegawa E, Hatazawa J, Abe K. Detection of neuroinflammation before selective neuronal loss appearance after mild focal ischemia using [18F]DPA-714 imaging. EJNMMI Res. 2018 Jun 8;8(1):43. doi: 10.1186/s13550-018-0400-x. PMID: 29884977; PMCID: PMC5993708.

    -Vicidomini C, Panico M, Greco A, Gargiulo S, Coda AR, Zannetti A, Gramanzini M, Roviello GN, Quarantelli M, Alfano B, Tavitian B, Dollé F, Salvatore M, Brunetti A, Pappatà S. In vivo imaging and characterization of [(18)F]DPA-714, a potential new TSPO ligand, in mouse brain and peripheral tissues using small-animal PET. Nucl Med Biol. 2015 Mar;42(3):309-16. doi: 10.1016/j.nucmedbio.2014.11.009. Epub 2014 Dec 6. PMID: 25537727.

    - In fig.1, please clarify what the chains R and R1 correspond to and insert in  “introduction” paragraph the reference to figure 1.

Round 2

Reviewer 1 Report

Authors revised their manuscript for each comment. I have only a minor comment.

Minor comments

L81. In “upregulation of miRNA-520 “h” expression”, “h” may be a typo.